# Stress, Coping and Considerations of Leaving the Profession—A Cross-Sectional Online Survey of Teachers and School Principals after Two Years of the Pandemic

**DOI:** 10.3390/ijerph192316122

**Published:** 2022-12-02

**Authors:** Petra Lücker, Anika Kästner, Arne Hannich, Lena Schmeyers, Janny Lücker, Wolfgang Hoffmann

**Affiliations:** Section Epidemiology of Health Care and Community Health, Institute for Community Medicine, University Medicine Greifswald, 17489 Greifswald, Germany

**Keywords:** teachers, school principals, perceived stress scale PSS-10, proactive coping: turnover, pandemic

## Abstract

Teaching is amongst the six professions with the highest stress levels and lowest job satisfaction, leading to a high turnover rate and teacher shortages. During the pandemic, teachers and school principals were confronted with new regulations and teaching methods. This study aims to examine post-pandemic stress levels, as well as resilience factors to proactively cope with stress and thoughts of leaving the profession among teachers and school principals. We used a cross-sectional online survey. The validated instruments Perceived Stress Scale (PSS-10) and Proactive Coping Subscale (PCI) were used. We included 471 teachers and 113 school principals in the analysis. Overall, respondents had a moderate stress level. During the pandemic, every fourth teacher (27.2%) and every third principal (32.7%) had serious thoughts of leaving the profession. More perceived helplessness (OR = 1.2, *p* < 0.001), less self-efficacy (OR = 0.8, *p* = 0.002), and poorer coping skills (OR = 0.96, *p* = 0.044) were associated with a higher likelihood of thoughts of leaving the profession for teachers, whereas for school principals, only higher perceived helplessness (OR = 1.2, *p* = 0.008) contributed significantly. To prevent further teacher attrition, teachers and school principals need support to decrease stress and increase their ability to cope.

## 1. Introduction

During the COVID-19 pandemic, preschools and schools were closed for weeks and months in many countries to contain SARS-CoV-2 outbreaks, with massive impacts on everyone’s day-to-day lives [1]. For instance, children were found to be less physically active, spent more time in front of screens, and have poorer eating habits [1]. Furthermore, it has been shown that the health-related quality of life of children and adolescents significantly decreased during the pandemic and mental health problems, such as anxiety and depression, increased [2]. So far, the effects of the COVID-19 pandemic have been predominantly studied among children, adolescents, and the elderly [2,3,4]. However, in terms of the school setting, teachers are just as important as pupils, as both groups benefit from each other’s good mental health [5].

To date, few studies examined the impact of the COVID-19 pandemic on teachers and school principals [6,7,8]. It was found that teachers experienced higher levels of stress than other occupational groups, which added to the already high levels of job-related stress, and that their quality of life decreased [9]. Those who worked remotely, reported more stress than those who taught in-person [10], which can be explained by the so-called ‘techno stress’, which includes being forced to expand abilities to use technology as well as working without social contacts [11]. In one study, 76.7% of the participants reported an increase in burnout during the pandemic [12], and in another, 58% had seriously considered leaving their job [13]. One study by Koestner et al. investigated the psychological burdens among teachers in Germany in March 2021 and found that symptoms of depression and generalized anxiety exceeded the level of the general population [14].

Even before the pandemic, teaching was shown to be one of the six occupations with the highest stress levels and lowest job satisfaction. This combination is associated with a high risk of burnout and turnover [15,16]. A pre-pandemic study from Germany showed that 95% of school principals enjoyed their work and 88% found it inspiring. Nevertheless, 53% of the participants reported chronic stress and 20% thought about changing their workplace [17]. Experiencing occupational stress means that both short-term and long-term mental and physical health may be negatively affected [18]. This is further aggravated by the fact that there is already a shortage of teachers in Germany, which will continue to get worse as a relatively large proportion of teachers will retire in the next few years. It is also estimated that about 10% of primary schools do not have a formally appointed school principal [19].

In Germany, in addition to their management and administrative function, most school principals have to teach a certain number of hours which gives them a dual role as leaders and teachers. In addition to their actual leadership and active teacher role, the pandemic has brought various problems and tasks: ever changing orders for infection control, which had to be interpreted, translated, and implemented [8], but also staff absences due to infections and quarantine while classes had to be maintained. Furthermore, the transition from face-to-face instruction to distance learning resulted in technical, organizational, and methodological challenges for many schools. In case of problems, principals are the contact people for pupils, parents, and staff as well as for superior agencies.

However, people react differently to potentially stressful situations. Whether a situation is perceived as stressful depends on the balance of the appraisal of a situation as demanding, and on sufficient available resources to cope with it [20]. Coping skills constitute a person’s resilience [21], which is the ability to recover, bounce back, adapt, or even thrive in the face of a challenging situation. It is a dynamic capacity [22], meaning that resilience can increase, decrease, and change over time. Different types of coping are described which can be adaptive or proactive [23]. Proactive coping is the ability to see stressful situations as a challenge, to grow because of them, and to continue to pursue one’s own goals [24].

The aim of this study was, therefore, to assess the prevalence and magnitude of stress, the ways of dealing with it, and turnover intentions among teachers and school principals after two years of the pandemic at the threshold of ‘normality’. Related questions, such as resources which contribute to resilience, were also addressed. In view of an existing teacher shortage [25] and an increased number of teachers who intend to leave the job compared to before the pandemic [26], it is important to know how stressed teachers are after these two years of pandemic, the most important causes of teachers’ stress, and how well equipped they are for future challenges.

## 2. Materials and Methods

### 2.1. Study Design

The online survey was carried out in Mecklenburg-Western Pomerania, a rural state in the north-east of Germany between 11 February and 7 March 2022. It was part of the project ‘schugi-MV’ (Scientific monitoring of school opening under pandemic conditions in Mecklenburg-Western Pomerania). Perceived stress and proactive coping abilities of teachers (including substitute teachers) and school principals of 615 schools were measured. Thus, a total of 615 school principals and 12,821 teachers [27] were approached.

### 2.2. Study Region

In Mecklenburg-Western Pomerania, the number of new infections with SARS-CoV-2 among 6- to 17-year-olds ranged from 277 to 1405 cases per day at the time of the survey [28]. The majority of pupils were in this age group. For teachers, there are no separate figures of infections publicly available. Infection control measures in schools were to wear masks and keep a distance throughout the school grounds, and for pupils, testing three times a week (rapid tests) was mandatory [29]. Teachers and principals had to enforce and ensure compliance with infection control measures which were rejected by part of the population.

### 2.3. The Questionnaire

The questionnaire was developed at the Institute of Community Medicine. Initially, we used Pubmed and Google Scholar for a semi-structured literature review with keywords related to (a) stress, burnout, (b) resilience, coping, (c) workplace violence, threat, (d) attrition, leave, turnover, job, and (e) teachers, school principals, and school to identify relevant articles. Standardized and previously validated instruments were used for the survey. Based on the findings of the literature review, further questions were added to the questionnaire, which were developed by the study team and have not been previously validated. The questionnaires for the participating groups were created in SoSci Survey [30].

### 2.4. Measurements

#### 2.4.1. PSS-10 (Perceived Stress Scale) [31,32]

The 10 questions, each with the 5-point Likert-scaled response options ‘never,’ ‘almost never,’ ‘sometimes,’ ‘quite often,’ and ‘very often’, refer to thoughts and feelings during the past month. Both positively and negatively polarized questions are included. For example, the stress level during the last month or the possibility of having had control over one’s own life are assessed.

The values of the individual items were summed up according to the developers’ specifications and can take on sum values from 0 to 40, with higher values indicating a higher stress experience. There are no standardized cut-off values; nevertheless, as in other studies (e.g., [33]), the following classification was applied: low stress experience (values between 0 and 13), moderate stress experience (values between 14 and 26), and high stress experience (values between 27 and 40). In addition to an overall assessment of perceived stress, the items form the two subscales ‘Perceived Helplessness’ and ‘Perceived Self-efficacy’. For perceived helplessness (six items), total scores from 0 to 24 can be achieved, with higher scores indicating higher levels of helplessness; for perceived self-efficacy (four items), total scores from 0 to 16 can be achieved, while higher scores indicate higher self-efficacy. Example items: ‘In the last month, how often did you have the impression that you were not up to all the tasks you have to do?’, ‘In the last month, how often have you felt confident about your ability to handle your personal problems?’.

The instrument has been widely used in different countries and showed a good reliability with Cronbach’s alpha values ranging between 0.74 and 0.91 [34]. Internal consistency of the overall scale in the current study is 0.90 for teachers and 0.89 for school principals. Cronbach’s alpha for both subscales and both groups ranges between 0.79 and 0.87.

#### 2.4.2. Proactive Coping Inventory (PCI), Subscale Proactive Coping [35,36]

The instrument applies a 4-point Likert scale with the response options ‘do not agree’, ‘hardly agree’, ‘rather agree’, and ‘agree exactly’ and comprises 17 items that relate to the constructs self-efficacy expectations, proactive attitude, and self-regulation as well as the assessment of stressful events as challenging. Both positively and negatively polarized questions are included. 

Values from the 17 items were summed up according to the developers’ specifications, yielding score values from 17 to 68. Higher values indicate better stress management. Example item: ‘I try to pinpoint what I need to succeed’.

Schwarzer et al. [37] reported a Cronbach’s alpha value of 0.86. The Cronbach’s alpha in the current study is 0.82 for both teachers and school principals.

#### 2.4.3. Further Variables

In addition to sociodemographic questions about age and gender, as well as the type of school where the teachers and school principals were employed, we provided the following questions (Table 1):

#### 2.4.4. Implementation of the Online Survey

E-mail invitations to complete the survey were sent to all schools in Mecklenburg-Western Pomerania by the State Ministry of Education and Day Care Facilities for Children. These were elementary (primary) schools (Grades 1–4), special schools (Grades 1–9), integrated/cooperative schools (Grades 5–12), regional schools (Grades 5–10), grammar schools (Grades 5–12 or 13), vocational schools, and sites with combinations of school types in the survey. The general university entrance qualification can be achieved at grammar and comprehensive schools upon completion of the 12th, respective 13th grade.

Additionally, the links was published on the project’s website. The school principals were asked to forward the link to the teachers. During the study period, one reminder was sent. The survey was conducted anonymously and does not allow any conclusions to be drawn about individuals. No IP addresses were stored. Participation in the survey was voluntary.

#### 2.4.5. Data Analysis

Statistical analysis was performed using SPSS Version 28. The standardized questionnaires were evaluated according to the developers’ specifications. If no answers were given for the PSS-10, the proactive coping scale, and whether leaving the profession was considered, the questionnaires were categorized as missing and were not included in the analyses. A further exclusion criterion was that the respondent’s age was not compatible with the questionnaire’s respective target group (i.e., most likely a wrong link was selected). In the case of missing information from individual questions, these were not considered in the presentation of the results.

Frequencies are given with absolute and relative numbers. For metric data, the mean is reported together with the standard deviation (for normally distributed data) or the median with the interquartile range (for non-normally distributed data and ordinal data). Nominal variables were analyzed using a Chi-square test, and ordinally scaled variables were analyzed using a Mann–Whitney U test. For the correlation between two at least ordinally scaled variables, Spearman correlation (with reporting of the rank correlation coefficient) was performed.

Logistic regression models were fitted separately for teachers and school principals to examine the factors that lead to the consideration of leaving the profession. We included the variables age, gender, perceived helplessness (total score), perceived self-efficacy (total score), proactive coping (total score), and feeling threatened as predictors/independent variables for thoughts of leaving the profession (Yes/No) in a binary logistic regression model. On the basis of a hypothesis-driven approach, each variable was first separately included in the model and then examined in the multivariable regression model. Odds ratios with the 95% confidence interval (CI) were presented. *p*-values <0.05 were considered statistically significant. The model’s goodness-of-fit was assessed using Cragg–Uhler’s (Nagelkerke’s) *R*^2^.

#### 2.4.6. Ethical Approval

The study was approved by the ethics committee of the University Medicine Greifswald (BB 163/21).

## 3. Results

Out of 693 questionnaires from teachers and school principals, we could include 584 questionnaires in the analysis. Response rates were 4.1% (teachers) and 23.1% (school principals). Characteristics of respondents are shown in Table 2.

### 3.1. Perceived Stress PSS-10 [33]

The results of the PSS-10 (Table 3) show that about two-thirds of 462 responding teachers (68.0%, *n* = 314) and two-thirds of 111 responding school principals (65.8%, *n* = 73) experienced a moderate stress level. Looking at the groups separately by gender, female teachers perceived significantly more stress compared to male teachers (17.2 vs. 20.4, *p* < 0.001). No gender differences were observed in principals regarding perceived stress (male: 17.7 vs. female: 18.6, *p* = 0.464).

#### Subscales Perceived Helplessness and Perceived Self-Efficacy

With possible scores ranging between 0 and 24 in the subscale ‘perceived helplessness’, in this study, mean values were 13.03 (*SD* = 4.45) for teachers and 12.62 (*SD* = 4.23) for school principals. Within the subscale ‘perceived self-efficacy’, values between 0 and 16 were achievable, and mean values for teachers were 9.24 (*SD* = 2.63) and 9.95 (*SD* = 2.51) for school principals.

### 3.2. Stressors

In response to the question ‘What is your biggest challenge in your schoolwork since the start of the Corona pandemic?’, 162 of the 467 participating teachers (34.7%) answered that it was the increased effort in teaching. For the 113 participating school principals, ever changing orders and measures were most often (36.3%, *n* = 41) a challenge. Teachers perceived the changing measures as demanding as the increased effort for lesson preparation (12.8%, *n* = 60). For both groups, the infection control measures in schools were often incomprehensible (teachers: 10.9%, *n* = 51, school principals: 15.9%, *n* = 18). Other aspects that were experienced as challenging were technical or internet problems (teachers: 7.7%, *n* = 36, school principals: 6.2%, *n* = 7). A lack of communication was a problem for 3.6% (*n* = 17) of the teachers and 9.7% (*n* = 11) of the school principals. For both groups, poor predictability of work (times, kind, mode) was a challenge (teachers: 8.6%, *n* = 40, school principals: 15.9%, *n* = 18), but not poor predictability of leisure time (teachers: 0.8%, *n* = 4, school principals: none).

More than 80% of the participating teachers (84.5%, *n* = 392) and over 90% of the school principals (91.0%, *n* = 101) reported that they did not manage to complete their tasks during the pandemic, or at least not to the extent that they would have liked to. For teachers, this relates more to the teaching of the subject matter and the individual support of pupils. School principals are not satisfied with the completion of development tasks. The tasks are shown in detail in Figure 1.

Another stressor was feeling threatened by pupils or parents when reinforcing infection control measures. More than one out of five teachers (22.0%, *n* = 103) and almost one in three school principals (31.3%, *n* = 35) reported having experienced such an incident at least once.

### 3.3. Proactive Coping (PCI—Subscale Proactive Coping)

Within this scale, values between 17 and 68 can be achieved. Higher values indicate a better ability to cope. Overall, there were no major differences between teachers and school principals in terms of their ability to deal with stress (Table 4).

Female and male participants did not differ significantly regarding proactive coping skills in both groups (teachers: *p* = 0.438, principals: *p* = 0.577). No differences between the age groups were found (*p* > 0.05).

Furthermore, we correlated the PSS-10 and its subscales with the PCI subscale ‘proactive coping’. Statistically significant correlations were found for all included variables (Table 5), whereby higher proactive coping values correlated with lower perceived stress and perceived helplessness and higher perceived self-efficacy in both groups. School principals showed stronger correlations between the variables than teachers. 

### 3.4. Resilience Factors

Being able to give multiple answers to the question ‘Who do you feel supported by during the pandemic?’, teachers (1083 answers) and school principals (283 answers) received the most support from their families, followed by people in the professional environment, which means by other teachers or school principals (teachers: 43.4%, *n* = 470, school principals: 42.4%, *n* = 120) (Figure 2).

For the question ‘What do you feel supported by and can draw strength from during the pandemic’ (multiple answers possible, teachers: 1191 answers, school principals: 273 answers), the greatest resilience factor in both groups was being with people who are important to oneself (teachers: 27.5%, *n* = 328, school principals: 27.8%, *n* = 76), followed by feeling valued (teachers: 20.0%, *n* = 237, school principals: 20.1%, *n* = 55) (Figure 3). The qualitative analysis for ‘Other’ revealed e.g. social contacts (teachers: 15.4%, *n* = 4, school principals: 50%, *n* = 3) and hobbies or being creative (teachers: 34.6%, *n* = 9, school principals: 16.7%, *n* = 1) as further resources for well-being.

Other sources of support are shown in Figure 3.

### 3.5. Consideration of Leaving the Profession

Serious consideration of leaving the profession since the beginning of the pandemic was reported by 27.2% (*n* = 127) of teachers and 32.7% (*n* = 37) of school principals (Table 6). Among the teachers, the largest share was contributed by those who work at vocational schools (24.4%, *n* = 31), among the school principals, it was those who work at primary (27.0%, *n* = 10) and special schools (27.0%, *n* = 10). There were no significant differences in teachers’ and school principals’ ages and gender distributions between those who are thinking about leaving the profession compared to those who are not considering this.

However, there were statistically significant differences regarding perceived stress, perceived helplessness, and perceived self-efficacy between those respondents who considered quitting their job and those who did not. Teachers and school principals who thought about leaving their job perceived more stress, had a higher score on helplessness, a lower self-efficacy, and felt threatened more often by pupils or parents. Teachers as well as school principals who considered leaving the profession showed lower proactive coping values; however, this did not reach statistical significance in school principals (*p* = 0.074).

### 3.6. Factors Associated with Consideration of Leaving the Job

Logistic regression models (teachers and school principals separately) were fitted to evaluate the factors which were associated with considerations of leaving the job (Table 7).

#### 3.6.1. Teachers

In the univariate model for teachers, perceived helplessness (OR = 1.26, *p* < 0.001), perceived self-efficacy (OR = 0.70, *p* < 0.001), proactive coping (OR = 0.94, *p* < 0.001), and a feeling of being threatened (OR = 2.00, *p* = 0.003) contributed to considerations of leaving the job. Age and gender showed no associations with these considerations in the univariate models. The goodness-of-fit of the multivariate logistic regression model for teachers was acceptable (Cragg–Uhler (Nagelkerke) *R*^2^ = 0.30) and showed the following results: Those teachers who felt more helpless (OR = 1.20, *p* < 0.001) were more likely to consider leaving the job. With higher values for perceived self-efficacy (OR = 0.81, *p* = 0.002) and proactive coping (OR = 0.96, *p* = 0.044), the probability decreased. After adjusting for perceived stress, coping abilities, age, and gender, feeling threatened did not further impact the chance of these thoughts (OR = 0.86, *p* = 0.594). Interestingly, although not significant in the univariable model, female teachers were less likely to consider leaving their job (OR = 0.37, *p* = 0.001) in the adjusted models.

#### 3.6.2. School Principals

For school principals, in the univariable model, perceived helplessness (OR = 1.32, *p* < 0.001), perceived self-efficacy (OR = 0.76, *p* = 0.002), and feeling threatened (OR = 2.88, *p* = 0.014) contributed to considerations about leaving the job. In the multivariable model only, perceived helplessness contributed significantly to thoughts of leaving the job—with a higher value compared to the univariate model. School principals were more likely to consider leaving their job (OR = 1.23, *p* = 0.008). Other factors such as perceived self-efficacy, proactive coping, and feeling threatened did not impact school principals’ considerations of leaving the profession. The overall fit of the multivariate logistic regression model was acceptable (Cragg–Uhler (Nagelkerke) *R*^2^ = 0.38).

## 4. Discussion

The aim of this study was to investigate the experience of stress among teachers and school principals after two years of the COVID-19 pandemic, their ability to cope, and considerations of leaving the job.

Although to varying degrees, both teachers and principals (who are always also teachers) showed effects of the pandemic regarding stress, coping, and considerations of leaving the profession. We were able to show moderate levels of stress among teachers and principals two years after the onset of the pandemic in our cross-sectional study, with one in four teachers and one in three principals considering leaving the profession. Increased perceived helplessness, lower perceived self-efficacy, and lower proactive coping skills were found to increase the likelihood of thoughts of leaving the profession among teachers, whereas only feelings of helplessness had a significant impact among school principals.

Most of the participating teachers and principals in our survey had a moderate stress level. As teachers and principals could look back on what they have achieved so far, one would rather assume at least an approximately ‘normal’ stress level among them. This is particularly noteworthy given that at the time of our survey, teachers and principals were already used to educating under pandemic conditions, most of them were vaccinated, infection levels were falling significantly, and most measures had already been lifted or were about to be lifted. Hence, it seems to take some time for the stress levels to normalize.

Publications on teachers’ stress during the pandemic, measured by the PSS-10, are scarce. However, it has been found that stress levels generally increased both in the population and among teachers. When applied to participants in community samples from 48 countries in the first three months of the pandemic in 2020, the PSS-10 showed a mean value of 17.4 [33]. Using the PSS-10-C, an instrument which was adapted to the COVID-19 pandemic, Oduacado et al. [38] found almost the same mean value (*M* = 17.27) in a community sample.

In 2021, health care professionals, an occupational group that was, due to the pandemic, also under more strain than before, had a mean value of the PSS-10 of *M* = 18.83 [39]. Compared to an Indian sample of teachers [40], the proportion of teachers with a moderate stress level in our study was similar (70.2% vs. 66.9% in our study), but in our sample, less teachers and principals had a high stress level (20.5% vs. 12.9% in our study). Women seem to experience generally more stress than men (e.g., [41]). In our study, this only applied to teachers. 

Similarly to us, Baker et al. [7] asked participating teachers what the most difficult aspect of their job was during the pandemic. Aspects such as ‘Lack of Connection’ (with pupils), ‘Online Teaching Challenges’, and ‘Increased Job Demands’ were also reported by our respondents.

One study by Alarcon found that the following factors, which most teachers faced during this time, were associated with teacher burnout [42]:Higher demands (e.g., distance teaching, infection control measurements, and the control of their compliance).Lower resources (e.g., lack of online study material, partially poor internet connection).Lower adaptive organizations (in Germany, schools are centrally regulated by the federal states).

These aspects were also highlighted as stressors in the survey in this study. In particular, the implementation of hygiene measures and lesson preparations represented stressors for teachers and school principals. For school principals, Fotheringham et al. suggest improving top-down communication. In particular, the communication of regulations and infection control measures for schools contributes to principals’ stress [8].

Individual support for students also required increased attentiveness as a result of the pandemic. These points should be addressed in order to be prepared for future challenges. Technical infrastructure and support as well as online learning material need to be improved—both for pupils and teachers. This would likely address most of the above-mentioned points as well as the aforementioned techno stress [11]. Another point is that due to the still ongoing pandemic, there are many infections among teachers and pupils. Teaching is therefore still restricted, so further monitoring of teachers’ and principals’ stress levels and sufficient support to protect and promote their well-being are needed. This would not only benefit teachers, but also pupils [5].

We also found that greater proactive coping skills correlated with lower perceived stress both among teachers and school principals. Studies using the PCI subscale during the pandemic were not available. Compared to participants in a pre-pandemic study by Mikus and Teoh, also from Germany, the level of proactive coping was somewhat lower among teachers in our study. A reason might be that the study by Mikus and Teoh [43] was conducted before the pandemic. In a study from the U.S., teachers reported higher levels of stress and lower ability to cope with stress after about a year of the pandemic than prior to the pandemic [44]. Overall, proactive coping skills contribute to teacher well-being and reduce burnout [43,45]. Resilient teachers are also better equipped to foster resilience in pupils [5], and therefore, teachers should be supported in strengthening their resilience. However, measures should be carefully balanced—by providing too many support opportunities at once (including for teaching), teachers can feel overwhelmed which then contributes to the experience of stress [46]. Just as important as acquiring skills to deal with stressors is that teachers and principals feel appreciated for what they have accomplished during the pandemic and beyond [19].

Apart from a resulting high workload, other factors contribute to the experience of stress in teachers and principals, including threats and violence. Although the media focus more on violence of teachers against pupils, a significant proportion of teachers report suffering various forms of violence by pupils [47] and parents [48]. This is an international problem [47] which affects teachers’ emotional and physical well-being, which in turn elevates the risk of burnout and turnover [49]. In a meta-analysis, Badenes-Ribera et al. [48] showed that teachers were more likely to experience non-physical than physical violence. Being threatened, as non-physical violence [50], describes something unpleasant or bad that someone says might happen [51]. In the literature, there is not much information on the prevalence of violence against teachers by pupils or parents. However, since teachers and principals had to enforce infection control measures that a section of society has rejected due to conspiracy theories [52], it is likely that they experienced resistance. In particular, the requirement for pupils to wear masks in school was viewed controversially. Teachers are more likely to experience non-physical violence, such as being threatened, than physical violence [48]. The adverse effect of workplace violence on job satisfaction and turnover intentions is known [53]. Promoting social-emotional learning would help teachers and school principals recognize and regulate both their own and others’ emotions. De-escalating strategies could help to prevent threats and violence from the beginning [48].

We were able to show in our study that, especially among teachers, increased stress levels and lower coping skills contributed to the consideration of leaving the job. Alarcon et al. also found that higher stress levels resulted in a more frequent staff turnover [42]. Interestingly, for both principals and teachers, the prevalence of thinking about leaving the profession doubled when they felt threatened at school, but after adjusting for stress and coping skills, this factor was no longer significant. This suggests that feeling threatened is also influenced by other predictors in the model, for example, perceived helplessness.

We cannot draw any conclusions about the impact of the pandemic on considerations of leaving the profession since we do not have longitudinal data from before the pandemic on teachers’ and school principals’ thoughts about leaving the workplace. However, the mere fact that in two years of pandemic, more than one in five teachers and about one in three principals have seriously considered leaving their jobs is alarming and even more critical considering the existing teacher shortage. In a study from the UK, the impact of the pandemic on the turnover intention could be seen: in January 2021, the number of teachers who considered leaving the profession almost doubled in comparison to before the pandemic [26].

Compared to a pre-pandemic study which included only school principals in Germany [17], in our study, the number of those principals who were thinking about leaving the profession was considerably higher (20% vs. 33%). In a study from the U.S. [54], principals’ intention to leave their job was correlated with teacher shortages, a problem that existed in Germany even before the pandemic and that will intensify in the coming years [19].

Thinking does not necessarily lead to actual action, but it is a clear indication of strong job dissatisfaction, which in turn is associated with burnout. Pupils whose mental health and well-being has been negatively affected by the pandemic [2] are encountering teachers who think about leaving the profession. It is to be feared that their learning success will be negatively affected in both the short and long term.

Therefore, the needs of teachers and school principals should be taken seriously and, in addition to improving the technical infrastructure at schools and solving organizational hurdles, additional programs should be initiated to provide support in terms of mental burdens. Higher self-efficacy may prevent teacher attrition by better mediating stress and thereby reducing the risk of burnout among teachers [55]. Mindfulness, for example, seems to have a positive impact on teachers’ stress [56]. Bonde et al. [57] were able to reduce mean PSS-10 scores by 2.1 points by providing mindfulness-based training in an intervention study in Denmark. 

Overall, the situation in German schools is unlikely to improve in the short term: the longed-for recovery from the pandemic during the summer months in 2022 was overshadowed by the war in Ukraine. Schools are already facing new challenges in teaching and integrating refugee children and dealing with the particular psychological needs of these pupils due to their experiences of war and flight [58,59]. These demands will further increase the risk of stress among teachers and principals. Teachers and principals need support in promoting their well-being and acquiring coping skills to be able to remain in the job.

### Limitations

One limitation is that compared to some other studies involving teachers [6], the response rate of teachers in our study was low. We assume that, as teachers and school principals had many additional responsibilities during the pandemic, they perceived the survey as another task and since participation was voluntary, many probably chose not to participate. However, the response rate among principals, who are always also teachers, was high (23.1%), and our results are comparable to those of other studies (e.g., [7,38]). Nevertheless, due to the cross-sectional design, the results represent only a snapshot, and we are unable to compare teachers’ and principals’ intention to leave the profession with the pre-pandemic situation. Furthermore, we provided different links for the different participating groups to access the questionnaires. It cannot be ruled out that someone has inadvertently or intentionally filled in another group’s questionnaire.

With regard to the measurement of stress experience and coping, there are a variety of measurement instruments available, some of which were first developed in the context of the pandemic. In contrast to other studies on this subject, we selected instruments for our survey that had already been implemented before the pandemic and that were sufficiently validated in national and international studies.

## 5. Conclusions

The impact and burdens of the COVID-19 pandemic have been studied to date with a strong focus on children and adolescents. The pandemic has also imposed heavy demands on teachers and school principals, such as implementing and maintaining hygiene measures, transitioning to distance learning (both technical and methodological), staff shortages due to illness, and addressing the increased needs of pupils. In the school setting, in particular, teachers, school principals, and pupils influence each other, and we therefore sought to examine the perceived stress, stressors, proactive coping abilities, and thoughts of leaving the profession of teachers and school principals two years after the onset of the pandemic in this cross-sectional study. We found moderate levels of stress in both teachers and school principals, which was inversely associated with proactive coping skills. Two years after the onset of the pandemic, one in three teachers and one in four school administrators had thoughts of leaving the profession, which were negatively influenced by increased perceptions of stress and lower coping skills in teachers, whereas in school principals, this was only intensified by feelings of helplessness.

In order to prevent further teacher and principal attrition, (post-pandemic) stress management programs need to be introduced, which should aim at reducing stress and at expanding coping skills and self-efficacy, and thus strengthening both teachers’ and principals’ resilience. It would be helpful to offer different programs to address group-specific needs. This is of paramount importance as, due to various factors, the pressures in the school setting are expected to increase further in the coming years.

## Figures and Tables

**Figure 1 ijerph-19-16122-f001:**
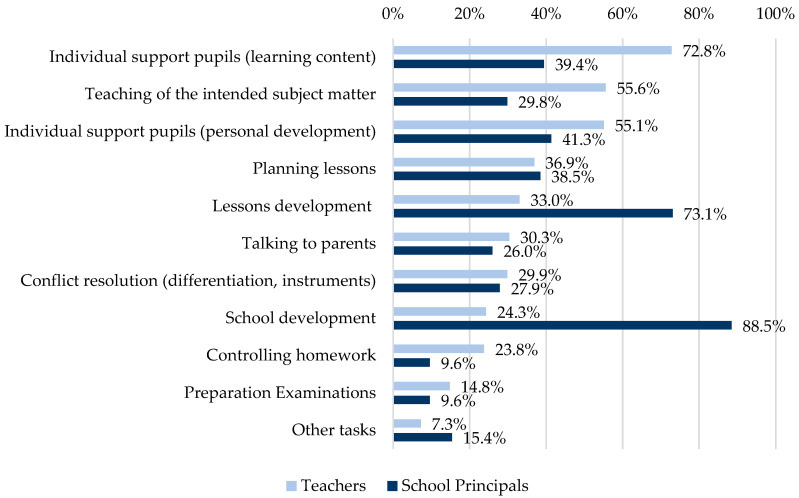
Which professional tasks do you not manage during the pandemic, or at least not to the extent that you would like to? (Multiple answers possible).

**Figure 2 ijerph-19-16122-f002:**
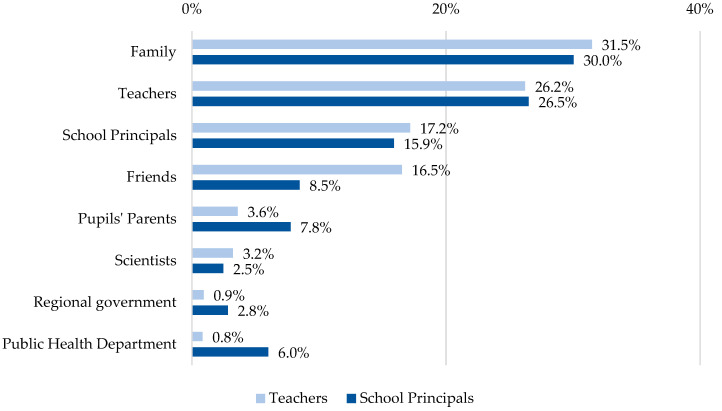
Who do you feel supported by during the pandemic? (Multiple answers possible).

**Figure 3 ijerph-19-16122-f003:**
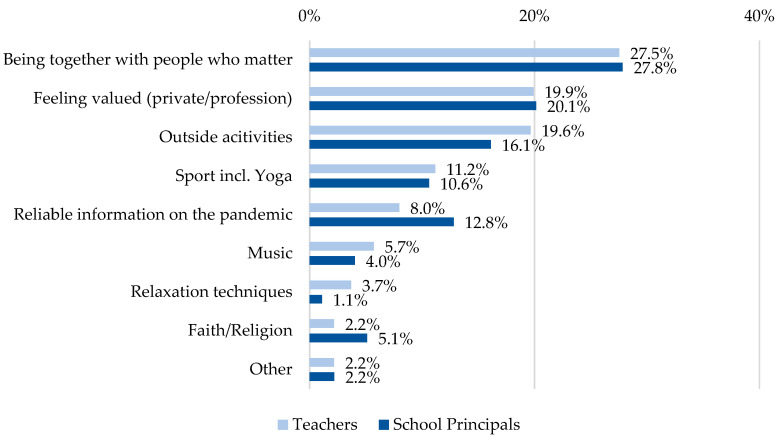
What do you feel supported by and can draw strength from during the pandemic? (Multiple answers possible).

**Table 1 ijerph-19-16122-t001:** Additional questions for teachers and principals.

Question	Answer Options
Who do you feel supported by during the pandemic? (Please select all that apply)	- Family - Friends - Pupils’ parents - School management - Colleagues/Teachers - Scientists - Health department - State government
What do you feel supported by during the pandemic? (Maximum 3 answers)	- To experience appreciation (private/professional) - Being with people who are important to you - Reliable information about the pandemic and everything that goes with it - Activities in nature - Sports incl. Yoga - Relaxation techniques - Faith/Religion - Music - Other, namely …
What has challenged you most about school since the Corona pandemic began? (Please provide only one answer)	- Increased effort for lesson preparation - Increased effort for teaching (e.g., parallel distance and face-to-face teaching in one class) - Problems with technology/IT/Internet - Difficulty in planning my work - Difficulty in planning my free time - Changing instructions/measures - Instructions/measures that are not comprehensible - Lack of communication (e.g., regarding current regulations) - Other, namely …
Are there any professional tasks that you can’t manage during the pandemic, or at least not the way you would like to?	- Yes - No
If you cannot manage everything, which tasks are left? (Please select all that apply)	- Lesson planning - Conflict resolution (differentiate, instruments) - Individual support of pupils in terms of learning content - Individual support of pupils in relation to their personal development - Control of homework - Teaching the intended subject matter - Exam preparation - Discussions with parents - School development - Classroom development - Other, namely …
Have you ever felt threatened by parents or pupils regarding the Corona hygiene measures in schools?	- Yes - No
Have you seriously thought about quitting your job since the pandemic started?	- Yes - No

**Table 2 ijerph-19-16122-t002:** Sample Characteristics.

	Teachers (*n* = 471)	School Principals (*n* = 113)
Gender, % (*n*) - female - male - diverse	362 (76.9) 107 (22.7) 2 (0.4)	77 (68.8) 35 (31.3) 0 (0.0)
Age (years), median (IQR)	47.0 (20)	52.8 (11.5)
Type of school, % (*n*) - Elementary school - Special school - Integrated/Cooperative school - Regional school - Grammar school - Vocational school - Site with combination of school types	57 (12.1) 26 (5.5) 59 (12.5) 81 (17.2) 114 (24.2) 114 (24.2) 20 (4.2)	29 (25.7) 19 (16.8) 12 (10.6) 15 (13.3) 17 (15.0) 16 (14.2) 5 (4.4)

**Table 3 ijerph-19-16122-t003:** Perceived Stress for Teachers and School Principals (PSS-10).

Perceived Stress Scale (PSS-10) (Possible Total Value 0–40)	Teachers *n* (%)			School Principals *n* (%)		
	Overall (*n* = 462)	Gender		Overall (*n* = 111)	Gender	
		Male (*n* = 107)	Female (*n* = 354)		Male (*n* = 34)	Female (*n* = 77)
*M* (*SD*)	19.6 (6.48)	17.2 (6.86)	20.4 (6.20)	18.3 (6.18)	17.7 (6.07)	18.6 (6.25)
Perceived stress - low - moderate - high	79 (17.1) 314 (68.0) 69 (14.9)	31 (29.0) 68 (63.6) 8 (7.5)	48 (13.6) 245 (69.2) 61 (17.2)	26 (23.4) 73 (65.8) 12 (10.8)	7 (20.6) 23 (67.6) 4 (11.8)	19 (24.7) 50 (64.9) 8 (10.4)

Note: *M* = Mean Value; *SD* = Standard Deviation.

**Table 4 ijerph-19-16122-t004:** Results Proactive Coping Scale.

Proactive Coping Scale M (SD)	Teachers			School Principals		
	Overall	Male	Female	Overall	Male	Female
	46.4 (6.57)	46.0 (6.48)	46.6 (6.59)	49.2 (5.86)	48.6 (7.32)	49.2 (5.10)
Age (years) - 20–39 - 40–59 - ≥60	46.14 (6.96) 46.17 (6.28) 47.14 (6.63)	47.3 (6.39) 45.0 (6.53) 44.2 (6.82)	47.05 (7.26) 46.43 (6.22) 46.67 (6.55)	48.50 (4.18) 49.24 (6.11) 48.91 (5.50)	48.8 (5.25) 49.4 (8.21) 46.2 (1.79)	48.3 (3.59) 49.2 (4.90) 49.7 (5.97)

Note: *M* = Mean, *SD* = Standard Deviation.

**Table 5 ijerph-19-16122-t005:** Correlations between proactive coping (PCI) and perceived stress, perceived helplessness, and perceived self-efficacy (PSS-10).

Proactive Coping Correlated with …	Teachers			School Principals		
	Spearman’s Rho	95%CI	*p*-Value	Spearman’s Rho	95%CI	*p*-Value
… Perceived Stress	−0.207	−0.298, −0.113	**<0.001**	−0.406	−0.559, −0.226	**<0.001**
… Perceived helplessness	−0.177	−0.270, −0.082	**<0.001**	−0.313	−0.481, −0.124	**0.001**
… Perceived Self-Efficacy	0.217	0.123, 0.308	**<0.001**	0.471	0.302, 0.612	**<0.001**

Note: CI = confidence interval.

**Table 6 ijerph-19-16122-t006:** Characteristics of teachers and school principals who considered leaving their job compared to those who did not.

Characteristics of Those Who Considered Leaving the Job Compared to Those Who Did Not	Teachers			School Principals		
	Considered Leaving Job (*n* = 127)	Did Not Consider Leaving Job (*n* = 340)	*p*-Value	Considered Leaving Job (*n* = 37)	Did Not Consider Leaving Job (*n* = 76)	*p*-Value
Age in years, *n* (%) - 20–39 - 40–59 - ≥60	39 (31.0) 70 (55.6) 17 (13.5)	107 (31.9 185 (55.2) 43 (12.8)	0.970 ^a^	2 (5.7) 27 (77.1) 6 (17.1)	6 (8.1) 49 (66.2) 19 (25.7)	0.510 ^a^
Gender, *n* (%) - male - female - diverse	35 (27.6) 92 (72.4) 0 (0)	72 (21.2) 267 (78.5) 1 (0.3)	0.292 ^a^	12 (33.3) 24 (66.7) -	23 (30.3) 53 (69.7) -	0.743 ^a^
Perceived stress, median (IQR)	24.0 (8.0)	18.0 (8.0)	**<0.001 ^b^**	23.0 (7.75)	16.0 (7.0)	**<0.001 ^b^**
Perceived helplessness, median (IQR)	16.0 (6.0)	12.0 (6.0)	**<0.001 ^b^**	15.5 (4.75)	11.0 (5.0)	**<0.001 ^b^**
Perceived self-efficacy, median (IQR)	7.0 (3.0)	10.0 (4.0)	**<0.001 ^b^**	9.0 (3.0)	11.0 (4.0)	**0.003 ^b^**
Proactive coping, median (IQR)	45.0 (8.0)	47.0 (9.0)	**0.001 ^b^**	48.0 (6.0)	50.0 (9.0)	0.074 ^b^
Felt threatened, *n* (%) - Yes No	40 (31.5) 87 (68.5)	63 (18.7) 274 (81.3)	**0.003 ^a^**	17 (47.2) 19 (52.8)	18 (23.7) 58 (76.3)	**0.012 ^a^**

Note: ^a^ = Chi²-test, ^b^ = Mann–Whitney U-test; *p*-values < 0.05 in bold.

**Table 7 ijerph-19-16122-t007:** Simple Logistic Regression (Dependent Variable ‘considered leaving the job’).

	Teachers (*n* = 427)				School Principals (*n* = 100)			
	Univariable Model		Multivariable Model		Univariable Model		Multivariable Model	
Variable	OR (95%CI)	*p*-Value	OR (95%CI)	*p*-Value	OR (95%CI)	*p*-Value	OR (95%CI)	*p*-Value
**Age**	1.002 (0.985, 1.020)	0.807	1.001 (0.979, 1.022)	0.990	0.992 (0.945, 1.042)	0.754	1.018 (0.953, 1.078)	0.592
**Gender** - **male** - **female**	ref. 0.709 (0.444, 1.132)	ref. 0.150	ref. 0.367 (0.198, 0.681)	ref. **0.001**	ref. 0.868 (0.372, 2.027)	ref. 0.743	ref. 0.699 (0.241, 2.028)	ref. 0.510
**Perceived helplessness (Sum 0 to 24)**	1.259 (1.186, 1.336)	**<0.001**	1.198 (1.106, 1.298)	**<0.001**	1.323 (1.164, 1.502)	**<0.001**	1.234 (1.056, 1.442)	**0.008**
**Perceived self-efficacy (Sum 0 to 16)**	0.700 (0.634, 0.773)	**<0.001**	0.805 (0.702, 0.923)	**0.002**	0.762 (0.637, 0.911)	**0.003**	0.848 (0.644, 1.117)	0.241
**Proactive Coping** **(Sum 17 to 68)**	0.941 (0.909, 0.973)	**<0.001**	0.961 (0.924, 0.999)	**0.044**	0.950 (0.885, 1.020)	0.160	1.001 (0.915, 1.096)	0.981
**Feeling of being threatened** - **No** - **Yes**	ref. 2.000 (1.257, 3.180)	ref. **0.003**	ref. 0.858 (0.488, 1.509)	ref. 0.594	ref. 2.883 (1.243, 6.687)	ref. **0.014**	ref. 0.609 (0.213, 1.742)	ref. 0.355

Note: OR = Odds Ratio, ref. = reference, statistically significant results in bold.

## Data Availability

The data analyzed during the current study are available from the corresponding author on reasonable request.

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
