# Peer review of "Stress, Coping and Considerations of Leaving the Profession—A Cross-Sectional Online Survey of Teachers and School Principals after Two Years of the Pandemic"

_ijerph, 2022, doi:10.3390/ijerph192316122_

Round 1
Reviewer 1 Report
Thank you for this opportunity to review your manuscript stress and teacher attrition two years into the COVID-19 pandemic. The pandemic has been an evolving crisis with many contours, and as such, it is important that we fully understand the impact it has had on teachers at several points throughout.
The instrumentation used in this study is appropriate given the aims of this research. The analytical approach was appropriate, and the inferences drawn were as well.
The one serious limitation to this work is the low response rate that you received from teachers, which you note at the beginning of the limitations section. Overall, I believe that this paper represents a good contribution to the larger body of knowledge about how teachers and school leaders have experienced the COVID-19 pandemic and its aftermath.
Author Response
Dear Reviewer 1,
Thank you very much for reviewing our manuscript and for your comments.
Kind regards
Reviewer 2 Report
Dear Author(s),
Thank you for submitting this interesting research work, which addresses a very relevant and timely topic.
Although many sections of the paper, i.e. methodology, results and discussions, are well addressed, other should be refined. In particular:
· A literature review section is missing.
· Conclusions should be extended. In particular, in the current version of the manuscript, I read the following sentence “Authors should discuss the results and how they can be interpreted from the perspective of previous studies and of the working hypotheses. The findings and their implications should be discussed in the broadest context possible. Future research directions may also be highlighted”. I agree with this insight, which can be useful for improving this section.
I suggest to carefully re-read the manuscript to correct any typos.
Best wishes!
Author Response
Dear Reviewer 2,
Thank you for your helpful comments.
A literature review section is missing.
We have added more information on the literature review, which included search terms relating to (a) stress, burnout, (b) resilience, coping (c) workplace violence, threat, (d) attrition, leave, job, turnover, and (e) teachers, school principals, and school. Page 3, Lines 103-107
Conclusions should be extended. In particular, in the current version of the manuscript, I read the following sentence “Authors should discuss the results and how they can be interpreted from the perspective of previous studies and of the working hypotheses. The findings and their implications should be discussed in the broadest context possible. Future research directions may also be highlighted”. I agree with this insight, which can be useful for improving this section.
We apologize for not having deleted this part from the template. The manuscript has now been carefully checked again. As suggested, we altered and extended the conclusions.
Kind regards
Reviewer 3 Report
The paper is interesting and well structured.
Authors should describe what was the situation like in Germany, where the data where collected. Which measures were adopted? in whcich way teachers and principals may have experienced threatens from parents/pupils?
Besides, data re analyzed ina na aggregated way; that is data collected from teachers were analized jointly with data collected from principals, exept for these referring to the "Consideration of leaving the profession". Whay they did so?
The perspectives of principals might be different from these of the teachres. This should be addressed/explained in the paper
Author Response
Dear Reviewer 3,
Thank you for your helpful comments.
Authors should describe what was the situation like in Germany, where the data where collected. Which measures were adopted? in whcich way teachers and principals may have experienced threatens from parents/pupils?
We have included (a) numbers of new infections per day among 6- to 17-year-olds and (b) a description of the infection control measures in place at schools at the time of the survey. These were a mask requirement and a distance requirement (where possible) for pupils and teachers throughout the school campus, and mandatory testing (rapid testing) three times per week. Furthermore, we have added another paragraph to the discussion, addressing the feeling of being threatened. Page 3, Lines 97-99
Besides, data re analyzed ina na aggregated way; that is data collected from teachers were analized jointly with data collected from principals, exept for these referring to the "Consideration of leaving the profession". Whay they did so?
Thank you for this comment. The groups were generally analyzed separately, with the respective results reported separately and shown in different colors. As the data for the logistic regression is more complex than the previous analyses, we chose to report the results for teachers and school principals separately.
The perspectives of principals might be different from these of the teachres. This should be addressed/explained in the paper.
We have added the following information: In Germany, additionally to their leadership role, school principals usually teach with a certain quota of hours. This places a double burden on them, but gives them also an insight into the roles of school principal and teacher. They are those who had to interpret, translate and implement changing infection control measures and who are the contact person in case of problems. Page 2, Lines 59-67
Although to varying degrees, both teachers and principals (who are always also teachers) showed effects of the pandemic regarding stress, coping and considerations of leaving the profession. These need to be addressed to promote their wellbeing and prevent a further attrition of both groups. For this reason, we believe it is appropriate to consider the two groups together. Discussion and Conclusion
Kind regards
Round 2
Reviewer 2 Report
The paper has consistently improved with respect to the previous version and can now be accepted in its current form. Congratulations!
Author Response
Dear Reviewer 2,
Thank you very much for reviewing our paper.
Kind regards